# Current Developments and Challenges in Plant Viral Diagnostics: A Systematic Review

**DOI:** 10.3390/v13030412

**Published:** 2021-03-05

**Authors:** Gajanan T. Mehetre, Vincent Vineeth Leo, Garima Singh, Antonina Sorokan, Igor Maksimov, Mukesh Kumar Yadav, Kalidas Upadhyaya, Abeer Hashem, Asma N. Alsaleh, Turki M. Dawoud, Khalid S. Almaary, Bhim Pratap Singh

**Affiliations:** 1Department of Biotechnology, Mizoram University, Aizawl, Mizoram 796004, India; gtmehetre@gmail.com (G.T.M.); vincentvineethleo@gmail.com (V.V.L.); 2Department of Botany, Pachhunga University College, Aizawl, Mizoram 796001, India; garima.singh106@gmail.com; 3Institute of Biochemistry and Genetics, Ufa Federal Research Center of the Russian Academy of Sciences, pr. Oktyabrya 71, 450054 Ufa, Russia; fourtyanns@googlemail.com (A.S.); igor.mak2011@yandex.ru (I.M.); 4Department of Biotechnology, Pachhunga University College, Aizawl, Mizoram 796001, India; mukiyadav@gmail.com; 5Department of Forestry, Mizoram University, Aizawl, Mizoram 796004, India; 6Botany and Microbiology Department, College of Science, King Saud University, P.O. Box. 2460, Riyadh 11451, Saudi Arabia; habeer@ksu.edu.sa (A.H.); asmalsaleh@ksu.edu.sa (A.N.A.); dturki@gmail.com (T.M.D.); Kalmaary@ksu.edu.sa (K.S.A.); 7Mycology and Plant Disease Survey Department, Plant Pathology Research Institute, ARC, Giza 12511, Egypt; 8Department of Agriculture and Environmental Sciences, National Institute of Food Technology Entrepreneurship & Management (NIFTEM), Industrial Estate, Kundli 131028, India

**Keywords:** plant viruses, diagnostics, next-generation sequencing, omics technologies, nanopore sequencing, quasi-species, in-field analysis

## Abstract

Plant viral diseases are the foremost threat to sustainable agriculture, leading to several billion dollars in losses every year. Many viruses infecting several crops have been described in the literature; however, new infectious viruses are emerging frequently through outbreaks. For the effective treatment and prevention of viral diseases, there is great demand for new techniques that can provide accurate identification on the causative agents. With the advancements in biochemical and molecular biology techniques, several diagnostic methods with improved sensitivity and specificity for the detection of prevalent and/or unknown plant viruses are being continuously developed. Currently, serological and nucleic acid methods are the most widely used for plant viral diagnosis. Nucleic acid-based techniques that amplify target DNA/RNA have been evolved with many variants. However, there is growing interest in developing techniques that can be based in real-time and thus facilitate in-field diagnosis. Next-generation sequencing (NGS)-based innovative methods have shown great potential to detect multiple viruses simultaneously; however, such techniques are in the preliminary stages in plant viral disease diagnostics. This review discusses the recent progress in the use of NGS-based techniques for the detection, diagnosis, and identification of plant viral diseases. New portable devices and technologies that could provide real-time analyses in a relatively short period of time are prime important for in-field diagnostics. Current development and application of such tools and techniques along with their potential limitations in plant virology are likewise discussed in detail.

## 1. Introduction

Viruses are the most genetically diverse organisms that cause infections in plants, animals, and humans. In general, viruses consist of a small genome that encodes only a few proteins, which makes it difficult to control viruses with a variety of methods [1]. In plants, among all the disease-causing pathogens, viruses present the substantial risk for agricultural production and have been reported to cause around half of the emergent infectious diseases among a variety of crop plants that cause about 40% of total crop losses upon viral infection [2]. Therefore, viruses represent one of the major constraints in agricultural production worldwide by decreasing both the quality and quantity of food crops [3]. Globally, more than twenty-five families of plant viruses are known to infect a variety of crop species, leading to higher economic losses [4]. Plant viral species are predominantly transmitted and spread by vertical and horizontal modes of transmission. In the vertical mode, the passage of the infectious agent occurs through the parent plants, either through vegetative propagation or by sexual reproduction via infected seeds. In the horizontal mode, spreading typically occurs through insect vectors, agricultural tools, as well as other direct or external forms of contamination [5]. Upon infection by a viral species, various symptoms, such as mosaic damage, yellowing, chlorosis, stunting, and necrosis are observed in infected plants, leading to loss of proper growth and reproductive functions of the plants [6].

Presently, very few treatments are available to manage or cure plant viral infectious diseases under field conditions. Therefore, early detection of the relevant causative agents represents the most crucial step to prevent the possible spread of the infectious disease [7]. The application of specific and sensitive methods for accurate detection is, therefore, required for efficient and effective plant disease management. In this context, innovative techniques and methods are continuously being developed and applied in viral diagnostics. However, most of the diagnostics methods used for viral detection and identification are expensive, time-consuming, and remain unsuitable for capturing emerging viruses. Nevertheless, there is a call to update these diagnostic methods for capturing new and emerging diverse viral species and their infections [8]. The most important difficulty with plant-based viral disease diagnostics is that plants are often infected simultaneously with different virus species or different variants of the same viral species [8,9]. In this context, the simultaneous detection of multiple viruses is the primary bottleneck in viral diagnostics. Moreover, due to the extraordinary genetic diversity among plant viruses, the use of one particular method for accurate detection becomes more complicated. Typically, viruses consist of genetic material, either RNA or DNA, with high mutation rates [10]. Currently, the preferred methods and techniques, such as electron microscopy (EM), immuno-serological methods (ELISA), molecular methods, and biosensor-based methods, are widely used to achieve rapid and reliable detection [11]. To increase the sensitivity and specificity, combinations of several techniques are predominantly employed in disease diagnosis. There is growing interest in assessing plant viral dynamics; however, effective strategies for the reliable detection of viruses in the infected plants were reported to be limited compared to those for human and animal viruses [12]. The recent developments in molecular biology and genomics techniques have enabled us to develop methods for plant viral diagnostics by obviating the need to culture viruses.

The advent of next-generation sequencing (NGS) technologies has invariably led to a revolution in plant virus diagnosis [13,14]. These methods do not require any previous knowledge of viral sequences and can sequence millions or billions of nucleotides in parallel, enabling the detection of all viruses, including those previously unreported [15]. Moreover, the use of such modern techniques to understand the ecology of the disease risk, emergence, and dynamics of viral diseases, alongside prior information about the virus and its diagnostics, is an emerging field of research [16,17]. In NGS-based methods, metagenomics is gaining more considerable attention for viral diagnostics. Various sequencing platforms, like those developed for second-generation and third-generation sequencing are mostly used for diagnostics. Second-generation sequencing platforms, however, need capital investments, and their high sample turnover rates are not very cost-effective [18]. Therefore, third-generation sequencing technologies such as the nanopore sequencing platform could provide valuable and affordable diagnostic tools for the accurate diagnostics of phyto-pathogens in crop health. In this review, we summarize the various techniques employed for plant viral diagnosis, with a special emphasis on the use of nanopore sequencing technologies (third-generation) in plant viral diagnostics.

## 2. Methods for Plant Viral Diagnostic

Numerous methods have been developed and commercialized for plant viral diagnostics. However, the practical application of each prevalent method depends on various factors, such as its cost, sensitivity, rapidity, the availability of instruments, and the stage of the disease [19]. Plants can display various physiological symptoms upon viral infection. Traditionally such visual inspections are used for viral disease diagnostics using indicator plants. Moreover, several other methods based on quantitative high-throughput imaging have been developed. With advances in molecular biology techniques, serological and nucleic acid-based methods are becoming essential tools in plant disease diagnosis. In recent decades, developments in NGS technologies have also provided a platform for viral diagnostics, and various viral species have been detected and identified by adopting these technologies [20]. A few of the most routinely used methods for the reliable detection of plant viruses are shown in Figure 1.

Extensive overviews of the important diagnostic methods, from traditional visual observations of the indicator plants to advanced third-generation sequencing technologies, are presented below.

### 2.1. Visual Inspections and Indicator Plants

Traditionally, visual inspections of infected plants and seeds were the most common method used for plant pathogen detection. At the beginning of the 20th century, symptoms of viral diseases affecting plants were proposed as a basis for the taxonomy of viruses [21]. The characterization of a plant viral disease with a known etiology usually depends on the symptoms expressed in the host plants. This is because of the easy identification of symptoms, especially when they appear to be disease-specific [22]. The external manifestations of viral infection of plants are closely associated with specific disorders in plant physiology, and the symptoms are divided into mosaics and chloroses. In the case of mosaics, changes occur in the colors and shapes of the leaves, while in chlorosis, systemic lesions are observed on the entire plant. In some cases, necrotic lesions are observed, which are additionally used for taxonomic identification of the virus—for example, viruses with necrotic mosaics and potatoes with streaks [23]. Symptoms of viral diseases include crinkling, browning of the leaf tissues, mosaics, and necrosis. Visual symptoms of viral diseases appear in a variety of forms. Depending on the species (variety) of the host and its resistance to the virus, primary diagnosis of the disease and the degree of damage to the plants can be determined [24]. Unfortunately, viral disease diagnosis using symptoms is more difficult than that for other pathogens due to the fact that several viruses can be present in a host that may alter symptom expression [25].

To measure the titter of infectious viral particles within infected plants, it is proposed to employ the method of plant infection and count the mosaic (necrotic) spots developed on the leaves of indicator plants depending on the concentration of viral particles, the number of spots that appear will vary. Empirically, it was noted, for example, that the tobacco mosaic virus titer can be investigated using isolated leaves of *Nicotiana glutinosa* L., *Datura stramonium* L., and *Phaseolus vulgaris* L. cultivated in a wet chamber [26]. Leaves of *P. vulgaris* were also suggested for the detection of tobacco necrosis virus and the ring spotting of tobacco, alfalfa mosaics, and common and southern bean mosaics. To identify and measure the concentration of potato viruses, it was proposed to use the leaves of *Gomphrena globosa* L. and white goosefoot *Chenopodium album* L. and, for cucumber mosaic, the leaves of *Vicia faba* L. (Dornburger variety) [27]. Young tobacco plants (*Nicotiana glutinosa*) can be used as an indicator of tomato aspermy virus, and *Gomphrena globosa* leaves can be used for the diagnostics of potato X-virus. These methods have many disadvantages, and their results are dependent on the ages of the leaves and the technique of rubbing the virus into the leaf. For a comparative analysis of these experiments, a method using leaf halves was proposed. The juice of control infected plants was applied to one half of the leaf, and the juice of another variant was applied to the other half. Subsequent staining of the infected leaves by iodine testing for starch, which shows externally invisible infectious zones (infected leaf cells produce a low concentration of starch), provides an improved modification that should also be considered [28].

In most cases, a visual inspection can be conducted by different methods, e.g., visible-light imaging, chlorophyll fluorescence imaging, hyperspectral imaging, and thermal imaging [29]. However, these methods are mostly influenced by various environmental and biological factors. Furthermore, visual inspections are not useful in asymptomatic cases [30]. Therefore, fluorescent proteins have remained effective investigative tools for deciphering biological processes and thus have been widely used as marker systems for the infectious process and the quantification of viral particles in host plants. The visualization of fluorescence tagged proteins in host plants using microscopic techniques was considered as one of the most efficient methods for viral diagnosis.

### 2.2. Microscopic Methods

One of the most classic methods for visualizing viruses in plant tissues is microscopic detection using modern light and high-resolution electron microscopes [31,32]. Certain viruses, after infection, lead to the formation of clusters of viral particles like inclusions or Ivanovsky crystals in the plant cells, which can be observed using light microscopes. Each type of virus assumes its own form of viral inclusions. Tobacco mosaic virus typically forms needle-shaped and hexagonal crystals, whereas Potato virus X and wheat mosaic virus are known to form spherical amorphous bodies [31,33]. Though microscopic methods were found useful for the detection of viruses, they need a high level of skill and expertise for the identification of viral species. Transmission electron microscopy (TEM) has improved the morphological characterization of viral particles in both crude and purified samples. TEM studies have resulted in one of the first proposals to classify viruses according to their characteristic groupings-based on their morphological and serological relationships, as well as some of their biological properties [34]. TEM provides the possibility of direct detection, where the infected tissue is initially homogenized followed by negative staining [35]. Various plant infecting viruses such as the Tomato yellow leaf curl disease (TYLCD), Potato virus S (PVS), Rice stripe mosaic virus (RSMV), Tomato brown rugose fruit virus (ToBRFV), Pepino mosaic virus (PepMV), and Potato virus M were observed and studied using electron microscopy techniques [36,37,38,39].

As the size and ultra-structural features are specific for each group of viruses, TEM could be used for reliable diagnosis as well as ultra-structural alterations during infections in plants. Zechmann and coworkers [40] reported TEM analysis with a microwave irradiation method for sample preparation to show ultra-structural alterations induced by TMV in *Nicotiana tabacum* and Zucchini yellow mosaic virus (ZYMV) in *Cucurbita pepo* plants in a relatively very short period of time. The authors demonstrated use of microwave fixation in TEM sample preparation could reduce time of investigation. EM has also been utilized to investigate subcellular localization of the plant virus in the vector to understand its transmission mechanisms. A study by Deng and colleagues [41] elucidated the transmission mechanisms of the Rice stripe virus (RSV), which is transmitted majorly by small brown plant hopper. TEM observation suggested that RSV might be replicating and accumulating in the inclusions of follicular cells and then spread into the oocytes. Alfalfa mosaic virus (AMV), also known as Potato calico virus, mostly leads to necrosis and yellow mosaic on a vast variety of plant species. The work on histological and cytological effects of AMV infection in potato leaf cells by implementing electron microscopy as well other techniques revealed significant ultra-structural changes [42,43]. Plants infected with AMV showed cytological changes in chloroplasts, mitochondria, cell membrane, and cytoplasm. The chloroplast become rounded, clumped together, and fragmented. In a similar ultra-structural study, Prune dwarf virus (PDV), an important viral pathogen of plum, sweet cherry, peach has been investigated to understand the intercellular transport mechanism of this virus. The authors observed structural changes in chloroplasts, mitochondria, and cellular membranes and also indicated that the virus transport through plasmodesmat avia MP-generated tubular structures [44]. Similarly, Rong et al. [45] investigated the ultra-structural mechanisms of Barley yellow dwarf virus-GAV (BYDV-GAV) in infected wheat plant using TEM. The study demonstrated yellow dwarf symptom formation in infected plant might be due to reduced chlorophyll content and chloroplasts fragmentation. In plant virology, TEM has reflected in elucidating structure of virus, its localization, and ultra-structure within host plant tissues. Therefore, in general, TEM mostly considered as the initial tools to urge the application of succeeding methods in identifying plant virus genus and species.

Many a times, other techniques such as serological are frequently combined with the electron microscopy to increase the detection sensitivity. In this context, immune electron microscopy (IEM) that basically uses specific antiserum against viral antigen has been developed. Several IEM variants like solid-phase immune electron microscopy (SPIEM) and or immunosorbent electron microscopy (ISEM) are primarily being used in viral diagnosis [46]. IEM can also be implemented directly using raw serum, avoiding the immunoglobulin purifications steps [35]. The microscopy techniques alongside serological methods are thus directly implemented to detect localized viral infections within infecting plants [38,47]. Bean common mosaic potyvirus (BCMV), an important seed borne pathogen, was detected using ISEM assay in young infected leaves of French beans [48]. Use of IEM has also been reported for yellow mosaic viruses (YMV) infecting legumes plant. The distribution of YMV in various parts of the seeds of naturally infected black gram (*Vigna mungo* L. *Hepper*) plant was determined [49]. Electron microscopy has played an important role in plant virology in elucidating its shape, size, and surface details for its identification and classification.

Another related application of microscopy employing a confocal microscope (CM) was used for detecting indicator compounds or molecules [50]. In this study, the CM was used to locate co-localized heat shock proteins like HSP70 and HSP90 in tomato plants that were infected by Tomato yellow leaf curl virus (TYLCV). CM has also been implemented for the detection of viruses labeled with fluorescent proteins, which enabled the tracking of viral particles in intracellular tissues of plants. Several researchers have used fluorescent viruses and/or plants that express green fluorescent protein (GFP) or red fluorescent protein (RFP) marker proteins [51]. These techniques enable one to observe the infectious process and quantification of the virus in plant tissues and to evaluate the natural or induced phyto immunity potential of the host plants, such as virus-induced gene silencing [52]. Moreover, the presence of a protein GFP or RFP allows one to observe the changes in infected plant tissues in real-time using modern fluorescent and confocal laser microscopes [53].

Currently, cryo-electron microscopy (cryo-EM) is the most advanced microscopy technique used in the investigation of plant-virus interactions, as this technique is capable of revealing nearly the atomic-level structures of viruses [35]. The cryo-EM technology has helped us understand structural conformations of viruses such as identifying certain nucleoproteins from negative-stranded RNA viruses, along with providing a better understanding of viral infections by illuminating the assembly and constituents involved in viral replication [54,55]. Advancements in microscopic techniques have thus helped us interpret the localization of viral infections through the detection of viral indicator molecules. In the future, further developments in these techniques may play a significant role in plant viral diagnosis and observations.

### 2.3. Serological Methods

Serological methods, such as ELISA (enzyme-linked immunosorbent assay), which are based on the reliable detection of a protein molecule using polyclonal or monoclonal antibodies, are widely used in plant viral diagnostics. Clark and Adams [56] were the first to utilize the ELISA method for diagnosing viral plant diseases, but it was earlier used to test the presence of viruses in plant tissues by serological methods [57]. ELISA-based methods, such as direct tissue blot immunoassay (DTBIA), double antibody sandwich (DAS) ELISA, and tissue-print (TP) ELISA are the most popular in viral identification [47,58,59]. Various modifications of ELISA were developed and used for the rapid detection of various plant viral diseases in various host plants. Among the various modifications of ELISA-based methods developed, the ACP-ELISA and antigen-coated plate enzyme-linked immunosorbent assay were used for RSMV identification in rice plants [37]. The Rice dwarf disease caused by Rice dwarf virus (RDV) was identified using purified RDV virions as the immunogen to prepare monoclonal antibodies.

Using a plate-trapped antigen enzyme-linked immunosorbent assay (PTA-ELISA), a dot enzyme-linked immunosorbent assay (dot-ELISA) was developed for the rapid detection of RDV in rice [60]. PTA-ELISA was reported to be useful for testing an adequate number of samples for the detection of viruses in a wide range of situations over a short period. A study showed the detection of various plant viruses, such as the Squash mosaic virus (SQMV), Cowpea severe mosaic virus (CPSMV), Cucumber mosaic virus (CMV), Cowpea aphid borne mosaic virus (CABMV), Zucchini yellow mosaic virus (ZYMV), and Papaya lethal yellowing virus (PLYV) in squash, cowpea, and papaya plants [61]. The detection of Tomato yellow leaf curl Thailand virus, (TYLCTHV) in tomato, pepper, eggplant, okra, and cucurbit plants was analyzed using a TAS-ELISA (triple antibody sandwich enzyme-linked immunosorbent assays) [62]. The Protein A sandwich enzyme-linked immunosorbent assay (PAS-ELISA) has been used for the detection of various *Badnaviruses* in West African yam plants [63]. Most of the modifications of ELISA were proven to be valuable tools for plant viral diseases due to their specificity; thus, these tools have been used for various crop plants (Table 1). Potato virus Y (PYV) represent a huge diversity in PVY strains and multiple recombinant virus genomes, therefore, fast and sensitive indexing techniques are requisite tools to determine the such virus at the early stages of viral infection. Enzyme-linked immunosorbent assay (ELISA) is therefore the most suitable methods for detecting PYV in potato and other susceptible crops such as pepper, tomato, and tobacco [64].

Significantly, these methods are able to detect even closely related strains of the same virus. However, there is a need for appropriate concentrations to detect viruses, where the reactions of enzyme-labeled antibodies function as the viral concentration, making the technique more applicable for the quantitative detection of viral load in the samples [70].

Another interesting improvement in serological methods was the development of immunostrips for the early and rapid detection of viral infections in plants. In this context, Byzova et al. [71] were instrumental in developing an immunochromatographic strip-based assay for the possible detection of five plant viruses (Bean mild mosaic virus, Spherical carnation mottle virus, Potato viruses X and Y, and Tobacco mosaic virus) inleaf extracts of virus-infected plants (sensitivity range 0.08–0.5 µg/mL for 10 min). The authors used multi membrane-based composites immobilized with polyclonal antibodies against the gold-conjugated antibodies and viruses in the developed assays. Recently, Niu et al. [72] used three similar immunochromatographic strip-based monoclonal antibodies (mAbs) to detect TZSV or Tomato zonate spot tospovirus. This type of strip-based detection was also reported to work with sandwich ELISA techniques for the detection of Citrus yellow vein clearing virus [73]. Interestingly, such developed strips were found to be capable of detecting the virus even at a sample dilution of 1:320 (w/v) and were also reported to be as effective as Dot-ELISA. The development of immunostrips allowed a considerable expansion of plant viral diagnostics directly in the field, as these strips can provide adequate results in a relatively short period. These strips are being actively implemented in crop production as an effective tool for monitoring viruses in seeds and plant materials. The most recent developments for the field detection of viral infections based on serological principles comparable to those of ELISA are being developed and marketed by Agdia Inc. https://orders.agdia.com/pathogen-tests/immunostrip-tests (accessed on 26 October 2020), with several immunostrips specific to various plant viruses, such as ImmunoComb^®^ strips for CMV, INSV, TMV, and TSWV and ImmunoStrip^®^ for the detection of Potato virus X and Potato virus Y (PVX and PVY).

### 2.4. Nucleic Acid-Based Methods

Nucleic acid-based methods are widely employed in many fields of diagnosis, including clinical, food safety, and environmental analysis. With the rapid development of molecular and genomics techniques, the scope of such techniques has become wider in the diagnosis of infectious pathogens. Due to their higher sensitivity, nucleic acid (DNA or RNA)-based techniques have been widely accepted in the field of viral diagnostics. In general, nucleic acid-based analysis has three essential steps: the isolation of the nucleic acid (either DNA or RNA), amplification, and product analysis. The last step is the most important, as it presents the result directly [39]. Various types of techniques have been developed for diagnosis using nucleic acids; however, most of these techniques are time-consuming and also need sophisticated instruments. Recent advancements in NGS techniques could provide a solution for in-field analysis, as well as analysis in limited resource areas. In the following, we discuss the most important techniques developed for plant viral analysis.

#### 2.4.1. PCR-Based Methods

Polymerase chain reaction (PCR) is one of the most important methods and is considered the gold standard for the molecular detection of various pathogens. Extensive modifications have been made for the PCR-based analysis of viral detection. PCR-based methods detect the DNA or RNA signatures of the virus using molecular primers to amplify a certain region of the viral genetic material and are among the most widely used methods for viral detection [74,75]. PCR-based methods using specific sets of degenerate primers are widely used for viral detection in various host plants. As a notable example, the tomato plant, which is cultivated worldwide and infected by several viral diseases like Tomato mosaic virus, Tomato leaf curl New Delhi virus, Tomato leaf curl Gujarat virus, Tomato leaf curl Palampur virus, as well as other viruses, was subject to viral detection based on the primer sets developed for these viruses. Conventional PCR is typically considered the most essential tool for the diagnosis of plant viruses (Table 2); however, due to the challenges associated with the optimizing conditions and the time needed for detection, these techniques remain less popular in plant viral diagnosis [76]. Additionally, conventional PCR is based on pure DNA that hampers its direct application in plant virology, as most of the plant viruses have an RNA-based genome. In this context, use of reverse transcriptase PCR (RT-PCR), which works on the principle of cDNA synthesis from RNA, is one of the most promising approaches in plant viral diagnosis [39]. Using RT-PCR, Nateqi and coworkers [77] identified Iris severe mosaic virus (ISMV) using specific primers. A one-step RT-PCR is more sensitive, reliable, and cost-effective and has been employed on a wide variety of the RNA genome-based viruses in plants. Li and colleagues [48] used one-step RT-PCR to detect Cherry flexiviruses such as Cherry green ring mottle virus (CGRMV) and Cherry necrotic rusty mottle virus (CNRMV), in *Prunus* species. Remarkably, authors used semi-automatic homogenizer for sample preparation, which make it ideal for screening substantial numbers of samples. In Kenya, for the first time, Tomato chlorosis virus (ToCV) was detected based on coat protein, minor coat protein, and heat shock protein 70 homolog genes specific primers using RT-PCR [78]. Use of the multiplex RT-PCR has equally been applied for detection of multiple viruses simultaneously. A study in virus-infected lily plant performed using simplex and multiplex RT-PCR with virus-specific primers demonstrated the detection of 3 different viruses, Cucumber mosaic virus (CMV), Lily mottle virus (LMoV), and Lily symptomless virus (LSV) in lily with mixed viral infections [79]. In the same way, Kumar and coworkers [80] showed simultaneous detection of five potato viruses including Potato aucuba mosaic virus (PAMV), Potato leafroll virus (PLRV), Potato virus M (PVM), Potato virus S (PVS), and Potato virus X (PVX). In pepino plant, Potato virus M (PVM), Pepino mosaic virus (PepMV), Tomato mosaic virus (ToMV), and Potato virus S (PVS) were detected using multiplex RT-PCR in China [81]. These and other such reports showed that multiplex RT-PCR demonstrated a huge potential in plant viral routine diagnosis.

Use of RT-PCR was also found valuable in analyzing the viruses like Pepino mosaic virus (PepMV), which has evolved as an emerging virus to tomato crops world-wide from the last few years. PepMV, a Potexvirus that gets transmitted mechanically and has assRNA-based genome. Various studies have successfully detected and identified different genotypes of PepMV in tomato crops. Using a newly developed one-step RT digital PCR, reliable quantification of different Pepino mosaic virus genotypes has been reported and found more suitable than only RT-PCR [82]. In another study, a RT-PCR was developed for the simultaneous detection and identification of three groups of Pepino mosaic virus (PepMV): European/Peruvian, Chilean 1/US1, and Chilean 2/US2 groups in routine analysis of field tomato samples [83]. Likewise, utilization of qPCR/real-time PCR for sensitive detection and quantification also reported for CTV in infected citrus plants and aphid vectors.

In the last decade, substantial developments in molecular methods that involve the implementation of multiplex detection methods have become cost-effective, as multiple viruses can be detected with increased efficiency. Using multiplex RT-PCR, it is possible to develop techniques to detect all members of a specific genus by designing new primers that target the conserved regions. Various viral genera like *Allexvirus* [99], *Begomovirus* [100], *Potyvirus* [101] were identified using multiplex RT-PCR techniques. Details on the multiplex methods used in plant viral diagnostics have been described elsewhere [102]. Possessing considerable importance in quantitative analysis during diagnostics, quantitative real-time multiplex PCR allows accurate detection and/or quantification at a low inocula load in samples. Real-time PCR technology additionally provides conclusive results by accurately discriminating between closely related organisms and is, therefore, used for the detection of a wide range of plant pathogens, including viruses and viroids [19]. There is much available information on relevant quantitative molecular biology methods, especially RT-PCR assays, which are widely used for a diverse range of plant viral disease diagnostics [75,102].

However, there are limitations in the detection of divergent variants of known viruses implementing these techniques. Therefore, refinement of PCR, RT-PCR-based techniques for more significant sensitive results are needed. Several groups were developing such techniques simultaneously from last decades. Among them, the methods based on the principle of coupling the sensitivity of ELISA and PCR were eventually termed immunocapture PCR or immunocapture RT-PCR (IC-PCR, IC-RT-CPR) were employed in detection of plant viral infections. In these techniques, the viral agents are captured by virus-specific antibodies bound to PCR tube, micro well plate which is followed by PCR or RT-PCR of the viral particles (Figure 2).

IC-PCR methods provide two-way specificity and were also found to be convenient for detection of viruses directly from crude plant extracts, without any requirement of the purified or partially purified virus [46]. Moreover, it can also avoid the need for nucleic acid (DNA or RNA) isolation. Therefore, such hybrid method which could provide double sensitivity is more versatile, and robust diagnostic techniques being successfully applied for plant virus detection, particularly in plant species or tissues that contain inhibitory substances [103]. Grapevine fanleaf virus (GFLV), a member of the genus Nepovirus was detected using a recombinant antibody to the GFLV coat protein (CP) by IC-RT-PCR [104]. In a study by Vigne et al. [105], a combination of serological and molecular methods with NGS was used for the detection and quantification of Grapevine fanleaf virus in vineyard samples. Immune capture PCR has been reported for various viruses and were found more robust, however, these methods need thermal cyclers that hinder their direct use under field conditions. Therefore, other methods like nucleic acid sequence-based amplification (NASBA), recombinase polymerase amplification (RPA), loop-mediated isothermal amplification (LAMP), cross-priming amplification (CPA), etc., have been developed as alternative techniques [39].

Due to its feasibility to work under field condition, LAMP has emerged as a robust, rapid and sensitive method for point-of-service detection and diagnosis of plant viruses. In general, LAMP works on mechanism of strand displacement and auto-cycling activity of Bst DNA polymerase, where amplification is done under isothermal conditions that do not require expensive devices like thermal cyclers [106]. Moreover, it also has additional advantage of direct observation of the results or by using some intercalating dyes. Various plant viruses like Barley yellow dwarf viruses [107], Cucumber mosaic virus [108], Potato leafroll virus [109], Southern tomato virus (STV) [110], Tomato brown rugose fruit virus (ToBRFV) [111], and Citrus leaf blotch virus in kiwifruit [112] have been detected using LAMP PCR and RT-PCR assays. Indian citrus ringspot virus (ICRSV) was analyzed using a novel, one-step RT-LAMP method by designing and testing four different primers that targets coat protein gene of ICRSV [113]. Warghane and colleagues [114] reported for the first time the development of RT-LAMP assays for rapid diagnosis of Indian Citrus tristeza virus (CTV) strains in citrus plant in India. Similarly, Melon necrotic spot virus (MNSV) has been diagnosed for the first time which causes significant economic losses in cucurbit crops using RT-LAMP. In their study, authors reported high specificity of RT-LAMP for MNSV without any cross-reaction with other viruses [115]. Eventually, different versions of LAMP are being developed for its more feasible use. Wilisiani and colleagues [116] developed LAMP assay with a portable device to detect begomoviruses under field condition. They have used a Genelyzer™ III portable fluorometer with a toothpick method and detected different members of begomovirus such as Tomato leaf curl New Delhi virus (ToLCNDV), Pepper yellow leaf curl Indonesia virus (PepYLCIV), and Tomato yellow leaf curl Kanchanaburi virus (TYLCKaV) in *Cucurbitaceae* and *Solanaceae* plants. The study clearly demonstrated use of the fluorometer portable device in developing the LAMP assay to achieve real-time detection of plant viruses under field condition. An Immunocapture version of the RT-LAMP ((IC-RT-LAMP) has been successfully applied in diagnosis of severe strains of Citrus tristeza virus (CTV) that cause quick decline and stem pitting in citrus [117]. In the past few years, LAMP techniques have a great effectiveness and the applicability in plant viral diagnosis even in poorly equipped laboratories with low resource settings due to its simplicity and ease in availability of required instruments [118]. Likewise, different variants of LAMP were developed and applied in diagnosis of different plant viral diseases and are receiving significant attention of plant virologist. However, major constraint of LAMP techniques is the design of the proper primer, which hinders their multiplexing application under field condition. Currently, NGS-based omics techniques have emerged as an innovative approach for virus discovery and detection. NGS technologies have proven very valuable in the detection of divergent variants of known viruses, as well as for detecting uncharacterized viruses that could not be found with existing detection procedures, such as PCR-and RT-PCR-based methods. The possible use of these current NGS-based methods for plant viral diagnostics and their sustainable future for in-field diagnostics are discussed below.

#### 2.4.2. Next-Generation Sequencing (omics)-Based Methods

Before the emergence of next-generation sequencing technologies, first-generation sequencing (Sanger sequencing) dominated the field of biological research. Even after the emergence of high-throughput sequencing technologies, these methods remained popular for analysis due to their higher throughput and relatively low cost [119]. NGS technologies have changed the approaches to both basic and applied research in plant virology. Due to the ability of NGS platforms to generate enormous data and their prompt delivery and cost-effectiveness, these techniques are becoming popular in various fields of biological research [120,121]. NGS-based techniques share a few familiar processes, such as the extraction of total nucleic acid (DNA or RNA) from the samples (environments, infected host plants, etc.), followed by fragmentation of the DNA for library preparation. By implementing a set of synthetic DNA adapters and primers to the fragmented DNA, one can use different sequencing chemistries and platforms for analysis. The term omics relates to various disciplines in biology like metagenomics, genomics, proteomics, metabolomics, etc., which explore the analysis of various cellular molecules through the advancement of NGS technologies. Among them, metagenomics represents the study of a collection of total DNA or RNA from a mixed community of organisms [122].

Metagenomics is being developed for the detection of viruses. In these approaches, total nucleic acid (DNA or RNA) is extracted from the infected plant samples and sequenced using NGS platforms, and viral sequences are identified using bioinformatics tools. Blawid et al. [123] described the details of sample preparation and the analytical steps required for the routine sequencing currently used for the discovery of plant viral sequences in metagenomics NGS data. In the past decade, viral metagenomics have explored plant virus biodiversity among crops and wild plants. A few of these studies have used a more ecological approach called “ecogenomics” that looks at the viral populations in individuals, rather than in the overall environment [15]. Viral metagenomics have been used to identify novel viruses in plants without bias, as this method does not require prior knowledge of the virus nor virus-specific primers. However, for previously unidentified viruses, due to the lack of reference sequences, these techniques employ de novo genome assembly, which is somewhat challenging [124]. The rapid use of metagenomics analysis in plant viral diagnosis is gaining much research interest. Various studies using metagenomics approaches (using both second and third-generation sequencing platforms) were reported for virus-infected plants. In the case of plants, most of the invading viruses are RNA viruses. Therefore, while isolating RNA, ribosomal RNA problematizes the identification of viruses. In this regard, ribosomal RNA depletion is implemented as the selection method to detect most of the viruses present in infected plants [125]. A study by Pooggin reported that RNA and DNA viruses, viral satellites, and viroids can be reconstructed to different degrees using deep sequencing and a bioinformatics analysis of small RNA populations from infected plants [126]. A brief account of the successful employment of both second and third-generation sequencing platforms in plant virology is discussed below.

#### 2.4.3. Analysis Using Second-Generation Sequencing Technologies

Second-generation sequencing technologies evolved based on high-throughput sequencing with the fast delivery of results at a reduced cost. Various platforms like GS FLX by 454 Life Sciences/Roche diagnostics; HiSeq, MiSeq, and NextSeq by Illumina, Inc.; SOLiD by ABI; and Ion Torrent PGM, Proton, S5, and BGISeq-500 are commercially available [127]. These sequencing platforms were found to be more sensitive and efficient in viral diagnosis compared to traditional approaches [128]. The detection of emerging and re-emerging viruses using second-generation sequencing technologies has evolved as the most promising approach in plant viral disease diagnostics [129]. Interestingly, these techniques were also found to be valuable for small RNAs (sRNA) from plants as a source to identify typical viral infections of the plants [130,131]. Plant viruses or viroids can also be detected indirectly using next-generation sequencing platforms. In virally infected plants, the host plant generates specific RNA molecules called short interfering RNAs (siRNA). RNA silencing (RNAi) is a cytoplasmic cell surveillance system used to recognize dsRNA and specifically destroy single and double-stranded RNA molecules homologous to the inducer using small interfering RNAs as a guide [132]. These methods provide significant opportunities to identify viruses or viroids infecting plants, even at extremely low titers, in symptomless infections and include previously unknown viruses or viroids [133]. NGS could identify a high number of siRNA sequences in the plant samples infected with viruses. The use of methods employing NGS techniques, mostly with second-generation platforms such as Illumina, in analyzing virus or viroid viromes in infected plant species was summarized by [20]. Using next-generation sequencing analysis, a novel gemini virus and a citrus chlorotic dwarf-associated virus were identified in a citrus plant [134]. These viruses are called Citrus leprosis virus cytoplasmic type 2 and the Citrus vein enation virus. NGS methods are a good option for investigating diseases of unknown etiology in plants. A novel *marafivirus* (Grapevine Syrah virus-1) was identified in vines affected bysyrah decline along with other RNA viruses infecting grapevine [20]. It was demonstrated that most of these second-generation sequencing platforms can reliably detect viral transcripts at frequencies of less than one in one million [135]. Furthermore, modern NGS techniques can detect multiple viruses simultaneously, along with the possible identification of hitherto unknown viruses [136].

Using Illumina MiSeq sequencing, Mumo and colleagues marked the genome sequences of four different viruses, a *potyvirus* (Moroccan watermelon mosaic virus) and *carlavirus*, including the Cowpea mild mottle virus (CpMMV) and two other putative *carlaviruses* related to Cucumber vein-clearing virus (CuVCV) from papaya leaves in Kenyan fields [137]. This study is the first report demonstrating the presence of these viruses in papaya in Kenya. The authors suggested that the Moroccan watermelon mosaic virus observed in papaya crops in Kenya indicates either the virus expanded geographical distribution or were previously undetected in papaya from Kenya. Viromes analysis of the infected plant samples was also found feasible for disease diagnostics. One such related study of grapevine analyzed the grapevine leafroll disease (GLD) caused by GLRaV-3 using the Illumina HiSeq 2500 platform. The study showed the presence of five viruses and three viroids in the infected vine. These viruses were Grapevine leafroll-associated virus 1 (GLRaV-1) and GLRaV-3 (genus *Ampelovirus*, family *Closteroviridae*) along with three viruses of the family *Betaflexiviridae* (namely, Grapevine virus A (GVA), Grapevine virus B, and Grapevine rupestris stem pitting-associated virus (GRSPaV). Crucially, this study clearly showed multiple distinct strains of three viruses (GLRaV-3, GVA, and GRSPaV) in the diseased grapevine [124]. Similarly, by adapting the Illumina MiSeq platform, deep sequencing of the sweet potato virus disease (SPVD) affecting sweet potato illustrated the genetic diversity of the viruses in both asymptomatic and symptomatic plant samples from South Africa. In this study, the authors showed that mixed virus infections of DNA and RNA viruses, *begomoviruses*, a *potyvirus*, and a *crinivirus* in individual plants could be identified without initial knowledge by employing Illumina sequencing platforms [138]. These results also indicated that in a particular disease, e.g., Sweet potato virus disease (SPVD), different viruses like SPCSV, SPFMV, and *begomoviruses* together play a significant role in development of the disease.

From the last decade, Illumina platform-based plant viral diagnosis has been applied to virome studies by analyzing the diseases of various important crops worldwide. Simultaneously, other available platforms like 454 (GS FLX) and Ion Torrent have been used for plant viral disease analysis (Table 3). However, each of these platforms has its own sequencing library preparation procedure, suitability for uses, and limitations for viral sequencing. Most of these platforms have been used in plant virology and were found to be most effective in viral diagnosis. In a virion-associated nucleic acid (VANA) analysis of the grain legume cowpea (*Vigna unguiculata* L.) using 454 (GS FLX), Palanga and colleagues identified three novel plant virus species (Cowpea aphid borne mosaic virus, Blackeye cowpea mosaic virus, and Cowpea mottle virus species) [139]. This study also indicated the presence of one novel *mycotymovirus*, which likely affects cowpea plants and associated fungal species from Burkina Faso, West Africa. Plant viral diagnoses were also reported using Ion Torrent PGM, which is a competitive technique in metagenomics due to its low sequencing cost and similarly efficient operation.

Using Ion Torrent, the genetic diversity of the Little cherry virus 1 (LChV1) in sweet and sour cherry species was studied [150]. This study highlighted the importance of genetic diversity among the LChV1 population, with the underlying implications of this diagnosis of viral agents included in cherry certification in several countries. Moreover, most of these techniques enabled the assembly of highly divergent viruses using whole-genome sequencing [162]. Luria et al. using modern sequencing techniques identified a new *tobamovirus* isolate in Israel that infects tomato varieties harboringTm-22 resistance grown in protected structures [163]. This unknown virus exhibits unique symptoms in tomato plants and is capable of infecting resistant pepper plants when cultivated on contaminated soil. Thus, high-throughput sequencing and bioinformatics analyses can be useful to analyze a considerable number of plant samples infected with numerous viral diseases. Various research groups have shown that these omics-based techniques could be used to detect viruses and viroids by sequencing both DNA and RNA viruses [13]. These techniques have developed at a rapid pace alongside contemporary developments in bioinformatics tools. Currently, third-generation sequencing techniques based on single-molecule analysis are gaining considerable research interest in every field of the biological sciences. These methods are used in plant viral diagnosis at the primary stage, but some research groups reported that these effective techniques may assist the field of plant virology.

#### 2.4.4. Analysis Using Third-Generation Sequencing Technologies

Second-generation sequencing transformed the field of biological research and is routinely used in environmental studies of microbial diversity, the human-associated microbiome, plant genomics, etc. During the last decade, third-generation sequencing technologies, also called single-molecule sequencing technologies, have gained much attention [164,165,166]. These techniques possess many advantages over second-generation sequencing technologies. For example, they can generate much longer reads from individual RNA or DNA samples, thus eliminating the need to assemble contigs de novo from short sequence reads. Additionally, these methods have very fast runtimes and, most importantly, require an extremely low input template (DNA or RNA), as well as provide real-time analysis. Presently, the Helicos™ Genetic Analysis System by SeqLL; LLC, USA, Single-molecule real-time (SMRT) sequencing by Pacific Biosciences (PacBio, USA), and MinION nanopore sequencing by Oxford Nanopore Technologies (ONT, UK) offer third-generation sequencing platforms [127].

The SMRT platform from PacBio has been used widely in various types of research as it can sequence single molecules. This platform uses hairpin adaptors to form a closed ssDNA template, SMRTbell, which is placed in a zeptoliter-sized chamber, with the zero-mode waveguide (ZMV) featuring an attached single polymerase molecule at the bottom of the chamber. With the system, the addition of fluorescent-labeled nucleotides in the phosphate group can be detected in real-time [167]. The SMRT platform has been employed for the diagnosis and identification of human viruses and bacterial pathogens; however, its application for plant viral diseases, such as fruit tree viral diagnosis, has not yet been reported [133]. Another leading platform is MinION nanopore sequencing from ONT. MinION nanopore sequencing is becoming more popular among researchers due to its simplicity and portability. MinION™ utilizesbase-specific fluctuations due to the blockage of a nanopore, ultimately transforming these fluctuations into DNA sequence information [168]. Oxford Nanopore devices perform direct DNA/RNA sequencing in real-time. For RNA viruses, MinION’s new direct RNA sequencing represents the next significant development; however, direct RNA sequencing studies are currently limited [169]. This technology is scalable from miniature devices to high-throughput installations like MinION, which was directly followed by GridION (designed to run five MinIONflowcells) and PromethION (designed to run 24 or 48 larger capacity flow cells), which use the same core technology as MinION but are designed for larger sequencing loads [119]. The pocket-sized MinION device from ONT is among the most popular devices for the real-time sequencing of DNA or RNA samples, as well as for whole genome sequencing [170]. A shorter MinION sequence involves skipping the cDNA synthesis step and is designed to directly ligate and sequence only polyadenylated RNA, thus providing strand-specific information and base modifications of RNA and demonstrating its use for RNA-based pathogens [171]. The considerable advantages of third-generation sequencing platforms such as MinION are their scalability via multiplexing and their long read lengths compared to other HTS platforms. Due to its long sequence read length, MioION provides a better phylogenetic resolution, which is a key metric for species-level identification and provides the ability to delineate species [172]. Worldwide, MinION nanopore technologies were found to be effective in delivering further insights and real-time analysis for a broad range of applications, such as epidemiological surveillance, microbiome identification, and field diagnostics [173,174]. In plant viral diagnosis, this technique can provide an effective and rapid method for the detection and identification of the causative agent of the disease. In the last few years, various studies have reported the effective use of nanopore sequencing in analyzing plant viral infections (Table 4). With the development of sophisticated bioinformatics tools, these techniques may change the current status of viral diagnosis in the field of plant virology, particularly in the areas of detection, identification, and genome sequencing.

Currently, Oxford Nanopore MinION has proven to be effective for real-time analyses with an average sequence length of more than 15 kb [168]. MinION has been used to detect and determine the whole genome sequences of a range of animal and human viruses, including the ebola33, dengue34, zika35, influenza31, cowpox36, and Ross River viruses; however, its use for plant viruses is still in the preliminary stages [173,174,175,176,177]. A few of the related studies have presented promising results using MinION nanopore sequencing. Various studies have shown that this technique is accurate and reduces the time needed for analysis; due to its portability, this method is being actively used to develop modern diagnostic methods for pests and diseases in agriculture. The application of MinION nanopore sequencing has been extended to plant microbiome characterization and pathogen identification in crop plants. At present, few studies have applied MinION nanopore sequencing for the detection or genomic analysis of plant viruses, such as Maize streak virus, Maize yellow mosaic virus, and Maize totivirus in maize crops or Plum pox virus (among others) (Table 4).

The portable nanopore sequencer has been reported to rapidly distinguish and accurately detect widely spreading plant viruses of the various agricultural plants. Potato virus Y (PVY) is the most economically important virus infecting cultivated potato (*Solanum tuberosum* L.), and high genetic diversity of this virus represents a challenge for its detection and classification. Della Bartola and colleagues [178] investigated Irish PVY isolates using conventional molecular and serological techniques. Additionally, nanopore sequencing was used to detect and reconstruct the whole genome sequence of four viruses (PVY, PVX, PVS, and PLRV) and five PVY genotypes in a subset of eight potato plants. Reconstruction of the genomes of PVY and other RNA viruses showed that nanopore technologies have enormous potential for virus detection in potato production systems and for the study of the genetic diversity of highly heterogeneous viruses like PVY [178]. This study demonstrated the identification of RNA viruses in potato plants using portable nanopore technology. The application of such a novel platform for the management of appropriate crop hygiene practices in commercial potato production systems shows the extended use of nanopore sequencing in the field of plant virology. Arabis mosaic virus (ArMV) is naturally transmitted to plants via nematodes in the soil. In a previous study, this virus was reported for the first time in a potato tuber. Given the importance of potato crops worldwide, the presence of ArMV in potato is rare [179]. Many *Begomovirus* species (family *Geminiviridae*) are known to cause economically important diseases in major crop plants. Nanopore sequencing has also been applied to detect, and for the genome sequencing of *Begomovirus* species. The single-strand DNA Sri Lankan cassava mosaic virus (SLCMV) was found to be a major emerging pathogen in Southeast Asia. Leiva and colleagues sequenced the complete genome of this virus using nanopore technology [180]. In the same way, a new bipartite *begomovirus* (family *Geminiviridae*) was also detected in cowpea (*Vigna unguiculata*) plants exhibiting bright golden mosaic symptoms on their leaves under field conditions in Brazil [183]. This study demonstrated that portable nanopore sequencing is effective for the rapid and accurate characterization of plant viral genomes.

Plant viruses are typically known to have expanded host range. The weeds of the *Solanaceae* family were observed to be important alternative hosts of *Begomovirus* (family *Geminiviridae*) in Brazil [189]. Tomato severe rugose virus (ToSRV) was identified inleaf samples of *Physalis angulata* using the rolling circle amplification method via ONT sequencing [182]. This study demonstrated that the natural infection of *P. angulata* by ToSRV makes one believeit has an expanding host range in tropical and subtropical weeds. Surprisingly, in a study of a wheat plant containing the Wheat streak mosaic virus (WSMV) resistance gene (*Wsm2*), the plant showed characteristic symptoms of WSMV. Serologically, WSMV was detected in all four samples. The infected plant samples were used to isolate RNA, and the single-strand cDNA was sequenced using Oxford Nanopore sequencing technology (ONT). In this study, nanopore technology confirmed the presence of WSMV and was also found effective for the detection of TriMV and BYDV in most of the samples. Deep sequence analysis of the single-strand WSMV revealed variation within the WSMV Sidney-81 reference strain, indicating new variants with Wsm2 resistance. The result of the study showed that nanopore technology can more accurately identify causal virus agents and has sufficient resolution to provide evidence of causal variants [184].

Most of the studies performed inplant viral diagnostics showed that Oxford Nanopore sequencing can remain a general method for diagnosis. To validate the use of nanopore sequencing for general diagnosis, Chalupowicz and colleagues infected the plants of several families with diverse pathogens like bacteria, viruses, fungi, and phytoplasma [186]. The total DNA or RNA was isolated from the infected plants along with seed samples for analysis with nanopore sequencing and data analysis tools. The result demonstrated the presence of pathogens in the infected plants; the authors detected these pathogens in real-time and with less than a one hour run time. The pathogens were classified to the species or genus level. Filloux and colleagues reported the detection and characterization of several plant viruses in yam plants using the MinION platform [187]. In this study, the authors presented efficient detection of Dioscorea bacilliform virus (DBV, *Badnavirus* genus), Yam mild mosaic virus (YMMV, *Potyvirus* genus), and Yam chlorotic necrosis virus (YCNV, *Macluravirus* genus) in the yam plants. The validation of the genome sequence of Yam mild mosaic virus obtained using MinION and the Sanger sequencing approach indicated that MinION could, in the near future, reliably enhance research on, and the monitoring of, plant viruses. Such studies reveal that third-generation sequencing technologies, especially MinION nanopore techniques, are becoming popular among plant virologists for effective viral detection and diagnosis. Due to its long read lengths, fast run times, portability, reasonable cost, and possibility to be used in every laboratory, the nanopore platform will, in the near future, benefit plant viral diagnosis for the early protection of the crop plants.

## 3. Current Challenges and Future Prospects (Nanopore Technology)

The management of plant viral diseases requires an effective diagnosis method that can generate early results for analyzing and preventing diseases. To date, various methods have been developed to identify and diagnose viral diseases among crop plants. The traditional diagnostic methods for plant viral detection, however, are time-consuming, bulky, and expensive, requiring cell culture and sample preparation. In recent years, sequencing technologies have been used in many fields, including metagenomics, medicine, plant physiology, and pathology. Therefore, advanced NGS-based techniques that offer large advantages in the diagnosis of viral diseases are becoming popular in analyzing the infectious diseases of bacterial, fungal, and viral species [190]. Furthermore, the current platform for NGS technologies provides rapid detection through high throughput analysis. To employ such tools, methods are being developed with high throughput and in a cost-effective manner. Several modern platforms and portable devices are becoming more readily available for investigations in the diagnosis of pathogens, such as viral, bacterial, and fungal pathogens that cause diseases in humans, animals, and plants [191]. The read lengths and reliability of such techniques allow us to investigate more deeply theviral genome diversity. Recently, researchers have highlighted the use of the portable DNA sequencing devices to study infectious organisms, such as bacteria and viruses [172]. Plant virologists are encouraged to use these technologies not only for viral diagnosis but also in the field of plant viral research. ONT is a third-generation sequencing technology that has many advantages. It has been successfully applied in a broad range of research areas, including human genetics, cancer, microbiology, plant, and infectious diseases [192,193]. ONT has been introduced as a portable MinION sequencer that serves the minimum needs for technical skills and bioinformatics knowledge. These factors make sequencing more feasible, allowing it to be performed under field conditions and in small laboratories [194].

However, there is a need for these revolutionary technologies to rapidly and accurately diagnose both identified and unidentified viral pathogens, as well as new variants of well-known pathogens [195]. Due to the limitations of a short sequencing length, several quasi-species that have importance in viral phylogeny and genetic variation remain understudied [196]. Therefore, the use of a consensus sequence to represent all information of a quasi-species is inaccurate. The Oxford Nanopore platforms offer the potential to study the genomes of specific viruses but still provide poor single-base accuracy. At present, the single-base accuracy of nanopore sequencers is around85%, and the accuracy of the corrected consensus sequence is 97% [197]. The Oxford Nanopore research group and many other researchers are working to improve the accuracy of sequencing results by utilizing nanopore equipment, reagents, and subsequent analytical algorithms. Likely, the on-site detection and diagnosis of emerging and re-emerging infectious diseases, based on nanopore sequencing technology, will represent a general trend in the future. As a prerequisite for future pathogen diagnostic methods, the developed techniques should have the ability to diagnose a substantial number of plant disease samples in less time and in a cost-effective way.

The development of innovative techniques should also satisfy anticipated needs more effectively than previous techniques. These techniques should be competitive in terms of their accuracy and cost-effectiveness and should involve all on-chip processing steps from sample preparation and enrichment to amplification. In this regard, the portable devices developed by Oxford Nanopore are more promising. Due to their ability to remain functional under an enormous range of operating conditions and their portability, these devices can be operated directly by growers and producers in the fields. Although sequencing techniques have experienced great advancements and are now able to provide a large number of datasets, there is still a need for the development of computer-intensive data processing steps like assembly, mapping, and alignment tools. The most fundamental problem in identifying novel viruses is associated with the requirements of iterative rounds of assembly and mapping with specialist knowledge of viral genomes. In this context, nanopore sequencing technology is a promising technology for viral tracriptome, structural variation, and genome re-sequencing studies, which can be performed directly in the field [198]. Nanopore sequencing is a rapidly maturing technology that delivers long reads in real-time and has been rapidly applied successfully in a variety of research applications. One key limitation of this technology is its high error rate, which, despite recent improvements to the nanopore chemistry and computational tools, still ranges between 5% and 15%. This reflects the low signal-to-noise ratio of ONT sequencing, which remains a key challenge. Several factors contribute to this problem, including the structural similarities between nucleotides and the multiple nucleotides concurrently influencing the signal [199]. ONT, therefore, developed as a flip–flop base-calling model that uses two overlapping windows to interpret the raw signal. Even though the output and error rates of the current generation of sequencers remain to be improved, new possibilities for evolutionary research using these novel techniques are expected to emerge [119].

More focused research on the development of such tools in the future will undoubtedly provide a more thorough understanding of the various components of viral pathosystems for future sustainable crop management. However, little progress has currently been made regarding the use of MinION sequencing in analyzing plant viruses. Possibly, most of these more innovative techniques that rely on the isolation of total DNA or RNA from the infected samples have difficulties in eliminating the host plant DNA or RNA, which makes sequencing more complex when interpreting the results. In a few of the cases, total double-stranded RNA (dsRNA) from pathogen-infected tissue was partially eliminated by hybridization to nucleic acid isolated from healthy plants to enrich the virus sequences in infected plant material before sequencing. Naito and colleagues detected the new bipartite *begomovirus* (family *Geminiviridae*) from cowpea (*Vigna unguiculata*) plants showing golden mosaic symptoms on leaves under field conditions in Brazil using MinION nanopore sequencing techniques [183]. The authors performed pairwise sequence comparisons with other *begomovirus* species previously reported infecting cowpea around the world which has resulted in low identity. Phylogenetic analysis revealed that the closest relatives with 85–87% nucleotide sequence identity were three legume-infecting *begomoviruses*; Macroptilium common mosaic virus, Macroptilium yellow vein virus, and bean golden mosaic virus [200].

Third-generation sequencers produce sequence reads with hitherto unprecedented lengths and will help to strongly increase the quality of genome assemblies. Further, the speed of sequencing and ease of sample preparation will facilitate sequencing in the field. The ONT device is pocket-sized, portable, and controlled by a laptop, thus offers immense advantage for in-field diagnostics. The first in-field use of ONT was demonstrated for human diseases during the 2014 Liberia ebola outbreak [201,202]. Therefore, these techniques directly provide a promising avenue for plant viral disease diagnosis. A putative workflow for the rapid and on-site diagnosis of plant viral infections using portable sequencing devices (MinION) developed by ONT is provided in Figure 3.

In this specific context, a case study was performed using an effective point-of-need field diagnostic system with MinION and MinIT mobile sequencing devices. In the field, Boykin and colleagues [203] developed the cassava virus action project (Tree Lab), which performed the in-field extraction and sequencing of cassava, an economically important crop and major food staple in sub-Saharan African countries (Tanzania, Uganda, and Kenya). The authors successfully established the on-site identification of viruses infecting cassava using MinION and MinIT mobile sequencing devices from ONT by developing an effective point-of-need field diagnostic system. The study demonstrated the successful detection and genome sequencing of cassava mosaic begomoviruses from the leaves, stems, and tubers of cassava plant and insect samples. Significantly, the whole analysis was performed within three hours starting from arrival on the farm to diagnosis and result interpretation. The study suggested that these newly developed tools enable the rapid diagnosis of plant viruses in the field and eliminate the need to ship samples to external sequencing service providers and centralized laboratories. With further developments in the relevant technological aspects for operation and their optimization under field conditions, such portable platforms will provide new avenues for the in-field diagnosis of plant viral infections and will gain greater attention from plant virologists in the future.

## 4. Conclusions

Monitoring plant health and quickly detecting pathogens are essential to reduce viral disease spread and facilitate effective management practices. Diagnostic methods commonly used to detect plant pathogens have limitations, such as the requirement of prior knowledge of the genome sequence, low sensitivity, and a restricted ability to detect several pathogens simultaneously. The development of advanced DNA sequencing technologies has enabled the determination of total nucleic acid content in biological samples. The possibility of using omics techniques, including third-generation sequencing platforms, such as Oxford Nanopores, as a general method for the diagnosis of plant diseases, is most readily applicable to viral diagnostics. However, there is still much to be discovered to advance the current state of viral diagnosis in plant diseases. Despite the great successes in the field of pathogen diagnosis tools, widely applicable, inexpensive, and simple approaches are still missing. Moreover, mixed infections remain a hurdle in disease diagnosis. The shortcomings need to be eliminated in all methods, including cell culture methods, molecular-based methods, immunological methods, and other advanced methods. Choosing the correct target site is vital for success in molecular detection, but finding such targets is not an easy task. Metagenomics has revealed many new viral entities recently and seems to be a promising approach. Further developments in bioinformatics tools using NGS methods could facilitate potential technological advancements in disease diagnosis that will help reduce crop losses due to emerging diseases.

## Figures and Tables

**Figure 1 viruses-13-00412-f001:**
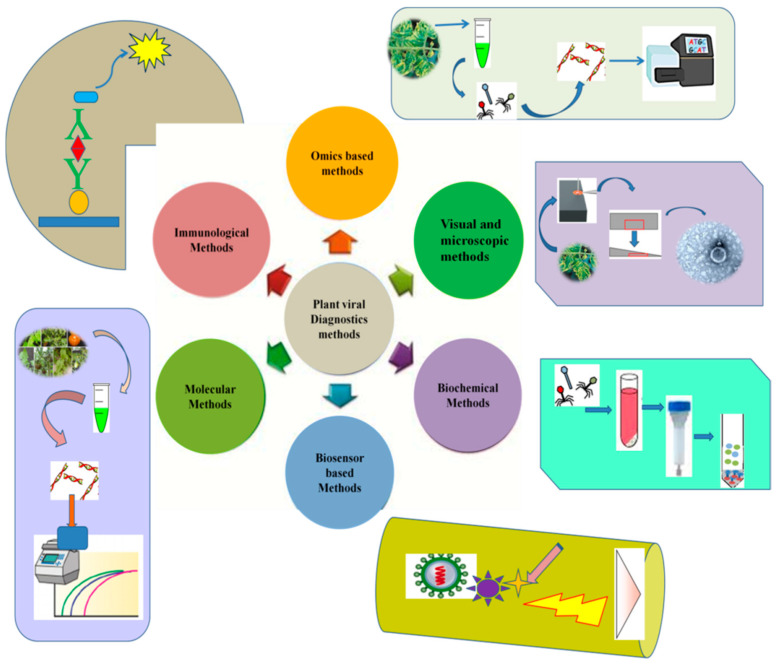
Common diagnostic methods used for plant viral identification and disease diagnosis of various crop plants.

**Figure 2 viruses-13-00412-f002:**
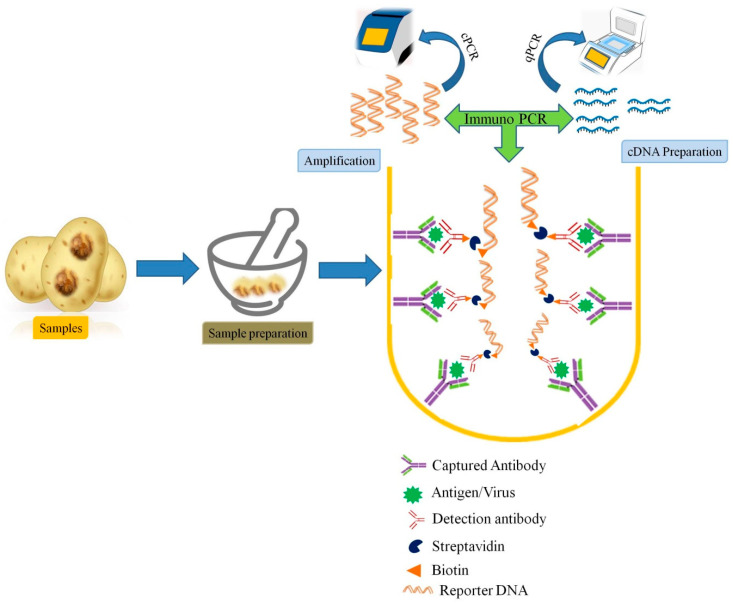
Schematic representation of Immunocapture PCR (IC-PCR) used in plant viral diagnosis.

**Figure 3 viruses-13-00412-f003:**
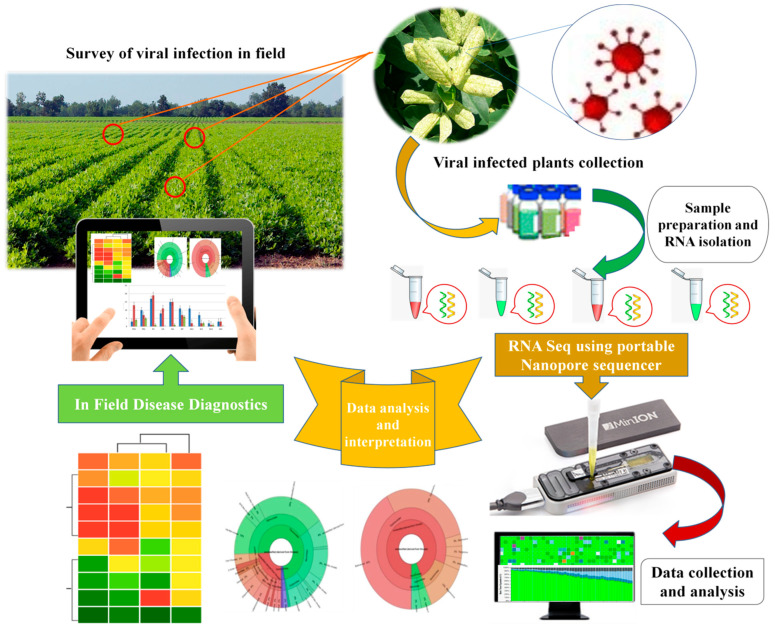
A putative workflow for rapid and on-site diagnosis of plant viral infections using portable sequencing device (MinION) developed by ONT, UK.

**Table 1 viruses-13-00412-t001:** Serological tests based on ELISA and its modified methods used employed in plant viral disease diagnosis for various important crops.

Virus Name	Host Plant	ELISA Based Method	Reference
Rice stripe mosaic virus (RSMV)	Rice	ACP-ELISA (antigen-coated plate enzyme-linked immunosorbent assay)	[37]
		Dot-ELISA	
		Tissue print-ELISA	
Rice dwarf virus (RDV)	Rice	PTA-ELISA (Plate-trapped antigen enzyme-linked immunosorbent assay)	[60]
		Dot-ELISA	
Squash mosaic virus (SQMV), Cowpea severe mosaic virus (CPSMV), Cucumber mosaic virus (CMV), Cowpea aphid borne mosaic virus (CABMV), Papaya lethal yellowing virus (PLYV)	Squash Cowpea Papaya	PTA-ELISA	[61]
Tomato yellow leaf curl Thailand virus (TYLCTHV)	Tomato, Pepper, Eggplant, Okra, Cucurbit	TAS-ELISA (triple antibody sandwich enzyme-linked immunosorbent assays)	[62]
Beet soil-borne virus (BSBV)	sugar beet	TAS-ELISA	[65]
Tomato yellow leaf curl Thailand virus, TYLCTHV	Tomato	TAS-ELISA	[62]
beet necrotic yellow vein virus (BNYVV)	sugar beet	TAS-ELISA	[66]
Potato virus Y (PVY)	Potato	TAS-ELISA	[64]
Potato virus S (PVS)	Potato	DAS-ELISA (double-antibody sandwich ELISA) direct ELISA	[36]
		Tissue print-ELISA	
		Dot-ELISA	
Rice-infecting viruses	Rice	DAS-ELISA	[67]
Badnaviruses	West African Yam	PAS-ELISA (Protein A-Sandwich ELISA)	[63]
Comovirus, Cowpea severe mosaic virus (CPSMV), Squash mosaic virus (SQMV)	Cowpea, Squash	IP-ELISA (Immune Precipitation ELISA)	[68] [69]

**Table 2 viruses-13-00412-t002:** List of conventional PCR and RT-PCR delineate primers targeting different conserved region of viral genome sequences for identifying viral infections.

Virus Name	Host Plants	Location	Primer Sequence	Reference
Pepino mosaic virus	Tomato	PepMV Ch1 genome	5′TTTGAGTATATCCCTGGTTC3′ 5′GTTAGCTAAACTACGTACAT3′	[84]
Pepino mosaic virus	Tomato	PepMV Ch2 genome	5′TTTGAGTATATCCCTGGTTC3′ 5′AATGGGATCCAAGCATTCCAG3′	
Tomato brown rugose fruit virus (ToBRFV)	Tomato	Tobamoviruses Universal primer	5′GGGAATCAGTTTCAAACRCA3′ 5′GGGGGGATTCGAACCYCT3′	[85]
tomato brown rugose fruit virus (ToBRFV)	Tomato	ToBRFV genome	5′GAAGTCCCGATGTCTGTAAGG3′ 5′GTGCCTACGGATGTGTATGA3′	[86]
Potato virus M (PVM)	Pepino	Coat protein	5′TAAGGTAAATCTGAAATAGTG3′ 5′GCCACCCTGGTTACGTGCTT3′	[81]
Pepino mosaic virus (PepMV)	Pepino	Coat protein	5′ATGCCTGACACAACACCTGT3′ 5′TTAAAGTTCAGGGGGTGCG3′	
Tomato mosaic virus (ToMV)	Pepino	Coat protein	5′ATGTCTTACTCAATCACTTC3′ 5′TTAAGATGCAGGTGCAGAGG3′	
Potato virus S (PVS)	Pepino	Partial Coat protein	5′ARCACCTTTAGGTTCACAG3′ 5′TTGARAAWCGCGAGTATG3′	
Tobacco mosaic virus (TMV)	Capsicum	Replicase gene	5′CGACATCAGCCGATGCAGC3′ 5′ACCGTTTTCGAACCGAGACT3′	[87]
Tomato mosaic virus (ToMV)	Tomato		5′CGAGAGGGGCAACAAACAT3′ 5′ACCTGTCTCCATCTCTTTGG3′;	
Tomato mosaic virus (ToMV)	Tomato, Tobacco	Genome sequence	5′GGGCCATATGGGGGCCA3′ 5′ATCCCCGGGT3′	[88]
Cucumber mosaic virus (CMV)	Cucurbit, pumpkin, snake gourd, bitter gourd	Coat protein	5′GAGTTCTTCCGCGTCCCGCT’3 5′AAACCTAGGAGATGGTTTCA’3	[89]
Papaya ringspot virus (PRSV)			5′GCAATGATAGARTCATGGGG’3 5′AAGCGGTGGCGCAGCCACACT’3	
Zucchini yellow mosaic virus (ZYMV)			5′ATAGCTGAGACAGCACT’3 5CGGCAGCRAAACGATAACCT3	
Cucumber Green Mottle Mosaic Virus (CGMMV)			5′TAAGCGGCATTCTAAACCTCCA’3 5′CACTATGCACTTTGGTGTGC’3	
*Begomovirus*			5′ATGKYGAAGCGACCAGCMGA’3 5′CGCCCKCMGAYTGGGMTTTTCTT’3	
Tobacco mosaic virus (TMV)	Tomato, Bell paper	Movement protein	5′TGAAAATGAATCATTGTCTG3′ 5′ACTCATCAACAACTTCTTCC3′	[90]
Tomato mosaic virus (ToMV)			5′TGAAAATGAATCATTGTCTA3′ 5′CATCTTCAATCAAATTATC3′	
Tomato leaf curl New Delhi virus (ToLCNDV)	Tomato, Eggplant, Cucurbits	Begomovirus universal primer	5′TAATATTACCKG WKGVCSC3′ 5′TGGACYTTRCAWGGBCCTTCACA3′	[91]
Tomato leaf curl Gujarat virus (ToLCGV)			5′TAATATTACCKG WKGVCSC3′ 5′TGGACYTTRCAWGGBCCTTCACA3′	
Tomato leaf curl Palampur virus (ToLCPalV)			5′TAATATTACCKG WKGVCSC3′ 5′TGGACYTTRCAWGGBCCTTCACA3′	
Tomato leaf curl Bangalore virus (ToLCBV)			5′TAATATTACCKG WKGVCSC3′ 5′TGGACYTTRCAWGGBCCTTCACA3′	
Tomato leaf curl New Delhi virus (ToLCNDV)	Tomato, Eggplant, Cucurbits	Begomovirus universal primer	5′ACNGGNAA RACNATGTGGGC3′ 5′GGNAARTHTGGATGGA3′	[1]
Tomato leaf curl Gujarat virus (ToLCGV)			5′ACNGGNAA RACNATGTGGGC3′ 5′GGNAARTHTGGATGGA3′	
Tomato leaf curl Palampur virus (ToLCPalV)			5′ACNGGNAA RACNATGTGGGC3′ 5′GGNAARTHTGGATGGA3′	
Tomato leaf curl Bangalorevirus (ToLCBV)			5′ACNGGNAA RACNATGTGGGC3′ 5′GGNAARTHTGGATGGA3′	
Tomato leaf curl New Delhi virus (ToLCNDV)	Tomato	Begomovirus coat protein	5′ATGKYGAAGCGACCAGCMGA3′ 5′CGCCCKCMGAYTGGGMTTTTCTT3′	[92]
Tomato leaf curl Gujarat virus (ToLCGV)			5′ATGKYGAAGCGACCAGCMGA3′ 5′CGCCCKCMGAYTGGGMTTTTCTT3′	
Tomato leaf curl Palampur virus (ToLCPalV)			5′ATGKYGAAGCGACCAGCMGA3′ 5′CGCCCKCMGAYTGGGMTTTTCTT3′	
Tomato leaf curl Bangalorevirus (ToLCBV)			5′ATGKYGAAGCGACCAGCMGA3′ 5′CGCCCKCMGAYTGGGMTTTTCTT3′	
Citrus tristeza virus (CTV)	Citrus	Coat protein	5′-CGCCATAACTCAAGTTGCG-3′ 5′-GTGGCCAATAGGTCCGTAGA-3′	[93]
Cucumber mosaic virus (CMV)	Tomato	Coat protein	5′GAGTTCTTCCGCGTCCCGCT3′ 5′AAACCTAGG AGATGGTTTCA3′	[94]
Tobacco mosaic virus (TMV)		Universal primer tobamovirus	5′ATTTAAGTGGAGGGAAAACCACT3′ 5′GTYGTTGATGAGTTCGTGGA3′	
Tobacco mosaic virus (TMV)		Cucumber green mottle mosaic virus	5′TAAG CGGCATTCTAAACCTCCA3′ 5′CACTATGCACTTTG GTGTGC3′	
Tomato mosaic virus (ToMV)		Universal primer tobamovirus	5′ATTTAAGTGGAGGGAAAACCACT3′ 5′GTYGTTGATGAGTTCGTGGA3′	
Tomato mosaic virus (ToMV)		Cucumber green mottle mosaic virus	5′TAAG CGGCATTCTAAACCTCCA3′ 5′CACTATGCACTTTG GTGTGC3′	
Potato virus Y (PVY)		Universal potyvirus primer	5′GGBAAYAATAGTGGNCAACC3′ 5′GGGGAGGTGCCGTTCTCDATRCACCA3′	
Potato virus Y (PVY)		Universal potyvirus primer	5′GTITGYGTIGAYGAYTTYAAYAA3′ 5′TCIACIACIGTIGAIGGYTGNCC3′	
Potato virus Y (PVY)		Papaya ringspot virus coat protein	5′GCAATGATAGARTC ATGGGG3′ 5′AAGCGGTGGCGCAGCCACACT3′	
		Zucchini yellow mosaic	5′ATAGCTGAGACA GCACT3′ 5′CGGCAGCRAAACGATAACCT3	
*Begomovirus*	Clove basil	Coat protein	5′ATGGCGAAGCGACCAG3′ 5′TTAATTTGTGACCGAATCAT3	[95]
*Potyvirus*			5′GTITGYGTIGAYGAYTTYAAYAA3′ 5′TCIACIACIGTIGAIGGYTGNCC3′	
Cucumber mosaic virus (CMV)			5′GCATTCTAGATGGACAAATCTGAATC3 5′GCATGGTACCTCAAACTGGGAGCAC3′	
Apple chlorotic leaf spot virus (ACLSV)	Pear	Coat protein	5′CCGAATTCATGGCAGCAGTTCTGAATC3′ 5′GAGAGCTCCTAGATGCAAAGATCAG3′	[96]
Apple stem pitting virus (ASPV)			5′ATGTCTGGAACCTCATGCTGCAA3′ 5′TTGGGATCAACTTTACTAAAAAGCATAA3′	
Tobacco mosaic virus (TMV)	Tomato	Coat protein	5′TCTTGTCATCAGCGTGGGC3′ 5′CCAGAGGTCCAAACCAAACCA3′	[97]
Cucumber mosaic virus (CMV)			5′TGGACAAATCTGAATCAACCAGTG3′ 5′GTACTAGCTCGTCCGTCTCG3′	
Potato virus X (PVX)			5′TCAGCACCAGCTAGCACAAC3′ 5′TGGTGGGAGAGTGACAACAGC3′	
Potato virus Y (PVY)			5′ATACTCGGGCAACTCAATCAC3′ 5′GCTTCTGCAACATCTGAGAAATG3′	
omato spotted wilt virus (TSWV)			5′CAGAATCTGGTAGCATTCAACTTCA3′ 5′ACTTTTCCTAAGGCTTCCCTG3′	
Tomato chlorosis virus (ToCV)			5′AGGGACCTCAGTTAAAGCAGC3′ 5′TCATGACTTCTGGCGTACCG3′	
*Tobamovirus*	Laboratory cultivated	Replicase gene	5′TKGAYGGNGTBCCNGGNTGYGG3′ 5′ACNGAVTBNABCTGTAATTGCTAT3′	[98]
			5′TKGAYGGIGTBCCIGGITGYGG3′ 5′ACIGAVTBIABCTGTAATTGCTAT3′	

**Table 3 viruses-13-00412-t003:** Next-generation sequencing (second-generation sequencing platforms)-based identification of causative agents associated with the viral diseases of various crops.

Host Plant	Disease	Viruses Identified/Findings	Platform Used	Reference
Papaya	Papaya ringspot disease	Moroccan watermelon mosaic virus (MWMV) Cowpea mild mottle virus (CpMMV) Papaya mottle-associated virus (PaMV)	Illumina MiSeq	[137]
Tomato	Begomovirus disease	Distinct ssDNA virus/subviral agents	Illumina HiSeq	[140]
China rose	China rose mosaic disease	Prunus necrotic ringspot virus (PNRSV)	Illumina HiSeq	[131]
Grapevine	Leafroll disease	Grapevine leafroll-associated virus 1 (GLRaV-1) and GLRaV-3	Illumina HiSeq	[124]
Grapevine	Grapevine rupestris vein feathering virus (GRVFV	Distinct GRVFV molecular variant	Illumina MiSeq	[141]
Tomato, Lettuce	Phytovirome	Virion-associated nucleic acids (VANA)	Illumina HiSeq	[142]
Sweet potato	Weet potato virus disease	Begomovirus, Potyvirus and Crinivirus	Illumina MiSeq	[138]
Common bean	Virus symptomatic bean plants	Pelargonium vein banding virus (PVBV) Bean common mosaic necrosis virus (BCMNVA) (Potyviridae), Aphid lethal paralysis virus (ALPV)	Illumina HiSeq	[143]
Cowpea	Virion-associated nucleic acids (VANA)	Bean common mosaic virus (BCMV) Cowpea mottle virus (CMV)	454 (GS FLX)	[139]
Malus, Prunus, Pyrus	Viruses and viriods	Genotypes of Apple stem pitting virus (ASPV)	Illumina HiSeq	[144]
Cucumber	Aphid lethal paralysis virus	Aphid lethal paralysis virus (ALPV)	Illumina MiSeq	[145]
Various plant species	Mixed viral infections	Viral/viroid species, putative novel viral species Cytorhabdovirus: CCyV1)	Illumina MiSeq	[146]
Common Bean	Bean common mosaic and necrosis disease	Bean common mosaic necrosis virus (BCMNV) Cucumber mosaic virus (CMV) Phaseolus vulgaris alphaendorna viruses 1 and 2 (PvEV1 and 2)	Illumina MiSeq	[147]
Common Bean	Virus symptomatic plants.	Pelargonium vein banding virus (PVBV) Bean common mosaic necrosis virus (BCMNVA)	Illumina HiSeq	[143]
Apricot	Mixed viral infections	Cherry virus A (CVA) Little cherry virus 1 (LChV-1)	Illumina HiSeq	[148]
Prunus	Various viral and viroids infections	Apricot latent virus (ApLV) Apricot vein clearing associated virus (AVCaV) Asian Prunus Virus 2 (APV2) Nectarine stem pitting-associated virus (NSPaV)	Illumina MiSeq	[149]
Sweet Cherry	Little cherry viral infection	Little cherry virus 1 (LChV1)	Ion Torrent	[150]
Maize	Maize lethal necrosis	Maize chlorotic mottle virus (MCMV) Sugarcane mosaic virus (SCMV), Wheat streak mosaic virus (WSMV) Johnson grass mosaic virus (JGMV).	Illumina MiSeq	[151]
Apple	Apple rubbery wood disease	Apple rubbery wood virus (ARWV) 1 and 2	Illumina HiSeq	[152]
Strawberry	Norovirus (NoV) gastroenteritis	*Norovirus* (NoV)	Illumina HiSeq	[153]
Peach, Prunus	Different disease phenotypes	Peach virus D, Viral synergisms involving genera in the Betaflexiviridae, Closteroviridae, and Luteoviridae families	Illumina HiSeq.	[154]
Tobacco	Different viral diseases	Cucumber mosaic virus (CMV) Potato virus Y (PVX) Tobacco mosaic virus (TMV) Tobacco vein banding Mosaic virus (TVBMV) Pepper mottle virus (PMV) Brassica yellow virus (BYV) Chilli venial mottle virus (CVMV)	Illumina HiSeq	[155]
Prunus	Prunus necrotic ringspot infection	Prunus necrotic ringspot virus (PNRSV)	Illumina MiSeq	[156]
Prunus	Mixed virus infections	Prunus necrotic ringspot virus (PNRSV) Prune dwarf virus (PDV) Apple mosaic virus (ApMV) American plum line pattern virus (APLPV) Ilarvirus-like RNA2 amplicon sequences	Illumina MiSeq	[157]
Grapevine	Virome of Grapewine	Grapevine fanleaf virus (GFLV)	Illumina HiSeq	[105]
Soybean	Pathogen analysis	Bean yellow mosaic virus (BYMV) and other pathogens	Illumina HiSeq	[158]
Grapevine	Genetic variability of infected viruses	Grapevine Syrah virus 1 (GSyV-1) Grapevine Cabernet Sauvignon reovirus (GCSV) Grapevine Red Globe virus (GRGV) Grapevine vein clearing virus (GVCV)	Illumina HiSeq	[159]
Grapevine	Mixed viral infections	Grapevine rupestris vein feathering virus (GRVFV) Grapevine yellow speckle viroid 1 (GSYVd-1)	Illumina HiSeq	[160]
Prunus	Stem-pitting symptoms	Marafivirus. Luteovirus-like viruses	Illumina HiSeq	[14]
Vanilla	Viral diseases	Potexvirus, Vanilla virus X (VVX) Vanilla latent virus (VLV)	454 pyrosequencing	[161]

**Table 4 viruses-13-00412-t004:** Next-generation sequencing (third-generation: MinION Nanopore)-based identification of causative agents associated with the viral diseases of various crops.

Virus Name	Host Plant	Remark	Reference
Potato virus Y (PVY)	Potato plant	Potato virus Y genetic diversity analyzed	[178]
Arabis mosaic virus (ArMV)	Potato plant	First report of ArMV in potato plants	[179]
Sri Lankan cassava mosaic virus (SLCMV)	Cassava plant	Complete genome sequencing of SLCMV	[180]
Sowthistle yellow vein virus (SYVV)	Sowthistle plant	Historically important plant rhabdovirus	[181]
Tomato severe rugose virus (ToSRV)	Tomato weed (*Physalis angulata* L.)	First report on infection of *P. angulata* by ToSRV	[182]
Cowpea bright yellow mosaic virus (CABMV)	Cowpea plant	Cowpea bright yellow mosaic virus identified	[183]
Wheat streak mosaic virus (WSMV)	Wheat plant	Wheat streak mosaic virus identified	[184]
Cassava mosaic disease (CMV)	Cassava plant	Cassava mosaic virus identified	[185]
Tomato yellow leaf curl virus (TYLCV)	Tomato plant	Tomato yellow leaf curl virus identified	[186]
Watermelon chlorotic stunt virus (WmCSV)	Watermelon plant	Chlorotic stunt virus identified	
Tomato brown rugose fruit virus (ToBRFV)	Tomato plant	Tomato brown rugose fruit virus identified	
Cucumber green mottle mosaic virus (CGMMV)	Tomato plant	Cucumber green mottle mosaic virus identified	
Zucchini yellow mosaic virus (ZYMV)	Butternut squash	Zucchini yellow mosaic virus	
Dioscorea bacilliform virus (DBV)	Water yam plant	Bacilliform virus identified	[187]
Yam mild mosaic virus (YMMV)		Mosaic virus identified	
Yam chlorotic necrosis virus (YCNMV)		Chlorotic necrosis virus identified	
Plum pox virus (PPV)	Prunus plant	Plum pox virus identified	[188]

## Data Availability

All data generated during this study are included in this article.

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
