# Peer review of "Current Developments and Challenges in Plant Viral Diagnostics: A Systematic Review"

_viruses, 2021, doi:10.3390/v13030412_

Round 1

Reviewer 1 Report

The paper presented for review contains a lot of information on virus detection. Unfortunately, it is very chaotic. Information is provided selectively. For example: “Over the years, many plant viral diseases, such as Tomato yellow leaf curl disease (TYLCD), Potato virus S (PVS), Rice stripe mosaic virus (RSMV), Tomato brown rugose fruit virus (ToBRFV), Pepino mosaic virus ( PepMV), and Potato virus M, have been documented using electron micrograph imaging. This is not the true. There are many more viruses that can be seen under an electron microscope.

Secondly, much of the information, including the first figures, is more relevant to a high school textbook than an impact factor journal. The presented graphics are completely unnecessary.

There is no point in presenting how DAS-ELISA works, everyone who works with viruses knows these basic diagnostic methods.

Table 1. Potato-virus Y (PVY)Potato plant TAS-ELISA[55].

Identification of PVY can be performed using the DAS-ELISA. But the TAS-ELISA is necessary, for example, for sugar beet viruses. There is no such information in the table.

Why are simple DAS-ELISA, DBIA tests drawn and there are no PCR reactions? This is more difficult and it would be much more interesting. Likewise, the PCR variants (e.g. IC-PCR) are not shown graphically

What should table 2 show? How were the viruses selected? Most viruses on tomato were listed, but PepMV was not mentioned.

2.4.1.PCR based methods

95% of plant viruses are RNA viruses. There is no information at presented paper about RT. qPCR, LAMP has been very briefly described. These are currently used diagnostic methods.

Lu, Y.; Yao, B.; Wang, G.; Hong, N. The detection of ACLSV and ASPV in pear plants by RT-LAMP assays. J. Virol. Methods 2018, 252, 80-85. LAMP was developed much earlier than in 2018. Why was this citation chosen? There are many previous publications on the use of the method in virus identification. It would be appropriate to refer to the first publications.

In my opinion, the work is to be rejected. It would make sense if the other proportions were kept. Two sentences about serological methods, two about TEM, and then in detail about qPCR, LAMP, NGS.

Author Response

Response to Reviewer 1 Comments

Dear Reviewer 1,

We would like to thank you for your valuable comments and suggestions, which helped to improve the manuscript. We have carefully revised the manuscript as per your and other suggestions. Please see below a response to the comments provided by you. (Changes incorporated in the revised manuscript are highlighted in yellow)

 Point 1:The paper presented for review contains a lot of information on virus detection. Unfortunately, it is very chaotic. Information is provided selectively. For example: “Over the years, many plant viral diseases, such as Tomato yellow leaf curl disease (TYLCD), Potato virus S (PVS), Rice stripe mosaic virus (RSMV), Tomato brown rugose fruit virus (ToBRFV), Pepino mosaic virus ( PepMV), and Potato virus M, have been documented using electron micrograph imaging. This is not true. There are many more viruses that can be seen under an electron microscope.

Response 1:Thank you for taking the time to assess our manuscript and for giving valuable suggestions.  We agree with you that many more plant viruses have been studied and documented using the electron microscope. Therefore, we have added few more examples citing the use of electron microscope for the detection or identification of plant viruses, page number 5 and 6, and line number from 193 to 237. Though, it would have been interesting to explore more on this aspect, however, it seems slightly out of scope, because, as per the theme of this special issue. Our focus is more towards the documentation of the use of next-generation sequencing techniques, especially third-generation sequencing technologies in plant viral disease diagnostics that has not been documented and discussed so much previously.

Point 2:Secondly, much of the information, including the first figures, is more relevant to a high school textbook than an impact factor journal. The presented graphics are completely unnecessary.

Response 2: Thank you for this suggestion. Since, we thought to give a brief schematic representation about the plant viral diseases, symptoms, and how these viruses are transmitted. As per your suggestion, we have removed figure 1 in the revised manuscript.

Point 3:There is no point in presenting how DAS-ELISA works; everyone who works with viruses knows these basic diagnostic methods.

Response 3:Thank you for this suggestion. As per your suggestion, we have removed figure 2 from the manuscript.

Point 4: Table 1. Potato-virus Y (PVY)Potato plant TAS-ELISA[55].

Identification of PVY can be performed using the DAS-ELISA. But the TAS-ELISA is necessary, for example, for sugar beet viruses. There is no such information in the table.

Response 4: Thank you for your suggestion. We have added more examples of using TAS-ELISA for viral identification for sugar beet viruses in table 1, row number 6-8.

Point 5:Why are simple DAS-ELISA, DBIA tests drawn and there are no PCR reactions? This is more difficult and it would be much more interesting. Likewise, the PCR variants (e.g. IC-PCR) are not shown graphically.

Response 5: We agree with you regarding this, therefore, we have removed figure 3 in the revised manuscript and added a new figure (Figure 2) showing the Schematic representation of Immunocapture PCR (IC-PCR) used in plant viral diagnosis. Also, we have discussed more on the use of Immunocapture PCR (IC-PCR) in plant virus diagnosis by citing more examples, page number11-12 and line number from 408- 435.

Point 6:What should table 2 show? How were the viruses selected? Most viruses on tomato were listed, but PepMV was not mentioned.

Response 6:Thank you for your comments. We have tried to explain the use of the conventional PCR method in the detection of plant viruses with the help of specific sets of degenerate primers designed by various research groups. As the list of such primers is very vast, we tried to enlist primers related to tomato viruses, as our work also focused on tomato crops. As per your suggestion, we have included the Pepino mosaic virus (PepMV) in the table (Raw 1, 2, and 3). Also, we have discussed more Pepino mosaic virus (PepMV) detection using RT-PCR in the text, page number 9, and line number from 378- 389.

Point 7: 2.4.1.PCR based methods

95% of plant viruses are RNA viruses. There is no information in the presented paper about RT. qPCR, LAMP has been very briefly described. These are currently used diagnostic methods.

Response 7: We agree with you regarding this, therefore, we have revised the manuscript with a more detailed discussion on the use of RT and qPCR in the revised manuscript, page number 8-9, line number 353- 377. A more detailed discussion about the use of LAMP methods for various plant viruses has been added in the revised manuscript, page number 13, line number from 439- 473. Although it would be interesting to discuss more these techniques, however, as per the special issue theme, we tried to discuss more the recent progress on utilizing NGS-based sequencing technologies in plant virus diagnosis.

Point 8:Lu, Y.; Yao, B.; Wang, G.; Hong, N. The detection of ACLSV and ASPV in pear plants by RT-LAMP assays. J. Virol. Methods 2018, 252, 80-85. LAMP was developed much earlier than in 2018. Why was this citation chosen? There are many previous publications on the use of the method in virus identification. It would be appropriate to refer to the first publications.

Response 8:Thank you for pointing this out. We agree with you regarding this, we have added the correct reference for the LAMP method “Notomi, T., Okayama, H., Masubuchi, H., Yonekawa, T., Watanabe, K., Amino, N., and Hase, T. 2000. Loop-mediated isothermal amplification of DNA. Nucleic Acids Res. 28, E63.”.page number 13,  line number 443 in the text, and page 28, line 1129, Reference number 110.

“Lu, Y.; Yao, B.; Wang, G.; Hong, N. The detection of ACLSV and ASPV in pear plants by RT-LAMP assays. J. Virol. Methods 2018, 252, 80-85” has been cited for primers of Apple chlorotic leaf spot virus(ACLSV) and Apple stem pitting virus(ASPV), page number 11, table 2, Reference number 99.

Point 9:In my opinion, the work is to be rejected. It would make sense if the other proportions were kept. Two sentences about serological methods, two about TEM, and then in detail about qPCR, LAMP, NGS.

Response 9: As per all of your suggestions, we have thoroughly revised the manuscript with the addition of more information about most of the diagnostic techniques such as electron microscopy (TEM), PCR based methods (RT-PCR, qPCR as well as for LAMP methods with citing more recent reports and information. Since the theme of this special issue is about the pros and cons of next-generation sequencing technologies in plant viral diagnosis, we embraced more towards the recent progress on utilizing NGS-based sequencing technologies for plant viral diagnosis. Next-generation sequencing (NGS) based innovative methods have shown great potential to detect multiple viruses simultaneously; however, such techniques are in the preliminary stages in plant viral disease diagnostics, especially the recently developed third-generation techniques such as Nanopore sequencing technologies. In recent years, Nanopore sequencing has been applied in research work related to human health, environmental microbiology due to its portability and high throughput analysis. Only in the last few years, use of Nanopore sequencing has been reported in the field of plant virology for diagnosis and virome analysis. In this review, we have discussed the work done by various research groups in plant viral diagnosis using the latest sequencing technologies such as MinIONNanopore technologies. As per our knowledge, this is the first review about the documentation of research work of application of Nanoporesequencing in plant viral diagnosis. We think that the use of such new portable devices and technologies that could provide real-time analyses in a relatively short period of time for in-field diagnostics could help early detection of plant viral diseases for their effective management.

Reviewer 2 Report

The manuscript reviewed the current developments and challenges in plant viral diagnostics, especially the recent progress on utilizing NGS-based sequencing technologies. The manuscript is well organized and well written. The tables in the manuscript are needed to improve further. Here are some suggestions:

1, the transmission mode is not the topic of this manuscript; it is not necessary to include the lower part of Figure1. Is there any particular reason to list these six virus-induced symptoms?

2, line 215, 'helped us understand structural conformations', whose structural conformation?

3, Figure 3, what does the DBIA stand for?

4, Line 240-242, spell the full name PTA-ELISA in the first place is enough.

5, all the tables need to improve further. E.g., name of virus or virus name, delete plant/plants in the second column of Table 1,Table 2. The serological test used, need to revise, ELISA-based methods? In Table 2, ‘gene-targeted’ is confusing; row 3-5 are primers not virus encoding sequences. The word ‘specific’ is not necessary. In Table 3, The column Host plant is needed to improve further. E.g., row 6 is ‘plants.’ In table 4, the first column should be the virus name.

Author Response

Response to Reviewer 2 Comments

Dear Reviewer 2,

We would like to thank you for your valuable comments and suggestions, which helped to improve the manuscript. We have carefully revised the manuscript as per your and other suggestions. Please see below a response to the comments provided by you. (Changes incorporated in the revised manuscript are highlighted in red)

The manuscript reviewed the current developments and challenges in plant viral diagnostics, especially the recent progress on utilizing NGS-based sequencing technologies. The manuscript is well organized and well written. The tables in the manuscript are needed to improve further. Here are some suggestions:

Response: Thank you for taking the time to assess our manuscript and for giving valuable suggestions. The manuscript has been revised as per your suggestions.

Point 1:the transmission mode is not the topic of this manuscript; it is not necessary to include the lower part of Figure1. Is there any particular reason to list these six virus-induced symptoms?

Response 1: Thank you for taking the time to assess our manuscript and for giving valuable suggestions. As per your suggestion, we have removed this part of figure 1. No sir, there was no particular reason to list these six virus-induced symptoms. We thought to give a brief schematic representation of the plant viral diseases, symptoms, and how these viruses are transmitted. However, after the removal of the lower part of figure 1, and also as per the suggestions received from the editor as well as the second reviewer, we have removed figure 1 completely in the revised manuscript.

 Point 2:line 215, 'helped us understand structural conformations', whose structural conformation?

Response 2: Thank you for taking the time to assess our manuscript, as per your suggestion, line 215 is revised (it was telling about virus confirmation), now page 6, line number 254.

 Point 3:Figure 3, what does the DBIA stand for?

Response 3: Thank you for taking the time to assess our manuscript. DBIA stand fordot-blot immunoassay or Dot-immunobinding assay (DBIA). However, an overall suggestion from the editor and second reviewer, figure 3 is removed from the manuscript.

 Point 4:Line 240-242, spell the full name PTA-ELISA in the first place is enough

Response 4: Thank you for your suggestion, as per your suggestion, only short-form (PTA-ELISA) has been used in line 242, now page 6, and line number 277.

Point 5:all the tables need to improve further. E.g., name of virus or virus name, delete plant/plants in the second column of Table 1, Table 2. The serological test used, need to revise, ELISA-based methods? In Table 2, ‘gene-targeted’ is confusing; rows 3-5 are primers, not virus encoding sequences. The word ‘specific’ is not necessary. In Table 3, The column Host plant is needed to improve further. E.g., row 6 is ‘plants.’ In table 4, the first column should be the virus name.

Response 5: Thank you for your valuable suggestions, all the table has been revised as per your suggestions.

Table 1 and Table 2: in both the tables, column 1 heading is written as ‘virus name’

Table 1 and Table 2: in both the tables, column 2 has been revised. The word plant/plants deleted.

Table 1: Serological test used is replaced with ‘ELISA-based method’.

Table 2: ‘gene-targeted’ is replaced with ‘Location’

Table 2: raw 3-5 are revised to the genome sequence.

Table 2: The word ‘specific’ is removed.

Tabl3 3: The column1 Host plant is improved, row 6 is revised.

Table 4: Column 1 heading replaced to ‘virus name’

Round 2

Reviewer 1 Report

In such a corrected form, the work looks much better. However, in my opinion there is still too much about serological methods and too little about molecular ones.